# Neural Sampling in Hierarchical Exponential-family Energy-based Models

**Xingsi Dong**[1,2,3]
dxs19980605@pku.edu.cn

**Si Wu**[1,2,3]
siwu@pku.edu.cn

1. PKU-Tsinghua Center for Life Sciences, Academy for Advanced Interdisciplinary Studies.
2. School of Psychological and Cognitive Sciences,
Beijing Key Laboratory of Behavior and Mental Health, Peking University.
3. IDG/McGovern Institute for Brain Research. Center of Quantitative Biology, Peking University.

## Abstract

Bayesian brain theory suggests that the brain employs generative models to understand the external world. The sampling-based perspective posits that the brain infers the posterior distribution through samples of stochastic neuronal responses. Additionally, the brain continually updates its generative model to approach the true distribution of the external world. In this study, we introduce the Hierarchical Exponential-family Energy-based (HEE) model, which captures the dynamics of inference and learning. In the HEE model, we decompose the partition function into individual layers and leverage a group of neurons with shorter time constants to sample the gradient of the decomposed normalization term. This allows our model to estimate the partition function and perform inference simultaneously, circumventing the negative phase encountered in conventional energy-based models (EBMs). As a result, the learning process is localized both in time and space, and the model is easy to converge. To match the brain's rapid computation, we demonstrate that neural adaptation can serve as a momentum term, significantly accelerating the inference process. On natural image datasets, our model exhibits representations akin to those observed in the biological visual system. Furthermore, for the machine learning community, our model can generate observations through joint or marginal generation. We show that marginal generation outperforms joint generation and achieves performance on par with other EBMs.

## 1 Introduction

Human behavioral studies [1, 2, 3] and animal neurophysiological studies [4, 5] have suggested that the brain performs statistically optimal Bayesian inference to interpret the external world [6, 7, 8, 9]. One promising theory for implementing Bayesian inference in the brain is to interpret the variability of neural responses as Monte Carlo sampling of the posterior distribution [10]. This perspective naturally accounts for the irregular firing patterns and other response properties observed in sensory cortex neurons [11, 12, 13]. Numerous sampling-based models [10, 14, 15, 16, 17, 12, 13, 18] have been proposed to elucidate neural dynamics and the underlying mechanisms of Bayesian inference.

To facilitate the brain's ability to derive meaningful representations from sensory input, it must continually update its generative model to approximate the true distribution of the external world [19]. Previous approaches often neglect this critical learning process. They either maintain fixed parameters for their generative model [15, 17, 16], or they employ biologically implausible methods like the variational approach with backpropagation (BP) for training [9]. Is there a generative model that integrates sampling-based inference with the capability to learn locally in both time and space?

37th Conference on Neural Information Processing Systems (NeurIPS 2023).

Energy-based models (EBMs) [20] provide a framework for inference with sampling method and learning with spatially localized rules. The reason for non-locality in time is the need to estimate the partition function. To estimate the global partition function, the network has to perform a top-down pass to obtain a negative sample. This process is referred to as the negative phase. During this pass, it disrupts the neural network's stored inference results, which is essentially the same reason for the temporal non-locality as in the case of BP. Recently, predictive coding networks (PCNs) [21, 22, 23, 24] use the Gaussian distribution to avoid the negative phase since the partition function of Gaussian is constant. But further study [25] reveals that the Gaussian assumption is restrictive when dealing with complex probability distributions. Moreover, setting aside biological constraints, estimating the partition function itself poses a significant challenge. In the field of machine learning, various sampling methods, such as amortized generation [26] and implicit generation [27], have been proposed to tackle this issue. However, both of these methods involve the use of the negative phase, which is known for being challenging to converge. Generative adversarial networks (GANs) address this issue by utilizing a discriminative model, but GANs are notoriously difficult to train in practice.

Besides, the inference dynamic of Gibbs sampling [28, 20] or Langevin sampling [29] in EBMs essentially perform random walks in local regions rather than the whole posterior space, which is too slow to be compatible with brain functions [16]. Therefore, it is crucial to investigate whether neural circuits in the brain have the capacity of realizing sampling-based inference rapidly.

**Summary of the work.** In Sec.2, we propose that our brain holds an EBM as the intrinsic generative model to interpret the external world. The neural dynamics employ a sampling-based method for Bayesian inference. Simultaneously, the learning dynamic aims to minimize the discrepancy between the observed distribution of intrinsic model and the real world. In Section 3, we introduce the Hierarchical Exponential-family Energy-based (HEE) model, which allocates the partition function across each layer. This allocation shifts the estimation of the total sample space required for calculating the partition function from a product of individual layer spaces to a sum of layer spaces. This approach enhances the convergence of our model. Furthermore, we efficiently sample the normalization term of the exponential-family in each layer using a group of neurons with fast dynamics. This localizes the learning process in both time and space. In Sec.4, we find that incorporating noisy adaptation, a generic feature of neuronal responses, into the inference dynamics effectively yields a second-order Langevin dynamic. In Sec.5, we validate the capabilities of the HEE model using *2D synthetic datasets* and *FashionMNIST* [30]. Then, we incorporate receptive field as the biological constrains to the HEE model training on *CIFAR10* [31]. We show that the HEE model can achieve a performance comparable with previous EBMs [27]. We also investigate the neural representation of semantic information, including orientation, color and category, which exhibit similarities to biological visual systems. And the neural adaptation can trigger neural phenomena including oscillations and transient which are widely observed in biological systems. In Sec.6, we discuss several related theories and models.

**Main Contributions.** We propose a hierarchical EBM whose learning process is localized both in time and space, which could potentially serve as a mechanism for the brain to utilize changes in synaptic strength for learning. And our brain-inspired EBM also presents a technique for estimating the partition function, which is a challenging problem within the machine learning community.

## 2  The intrinsic generative model

In this section, we propose that our brain holds energy-based models (EBMs) as the intrinsic generative model consisting of two components: inference and learning. Inference is believed to be carried out through neural sampling [10, 20], while learning is accomplished through long-term synaptic plasticity [32].

Let $\mathbf{x}$ be the observation received by our brain, and let $\mathbf{z}$ be the latent variable represented by neurons. The joint distribution of the EBMs is written as $p_\theta(\mathbf{x}, \mathbf{z}) = p_\theta(\mathbf{x}|\mathbf{z})p_\theta(\mathbf{z})$, where $\theta$ are stored in the connection weights of neurons (Fig.1A). The EBMs aims to minimize the difference between the intrinsic marginal distribution $p_\theta(\mathbf{x})$ and the true distribution of the external world $p_{\text{true}}(\mathbf{x})$, which is described by the Kullback–Leibler divergence,

$$\min_\theta D_{\text{KL}} \left[ p_{\text{true}}(\mathbf{x}) \parallel p_\theta(\mathbf{x}) \right]. \tag{1}$$

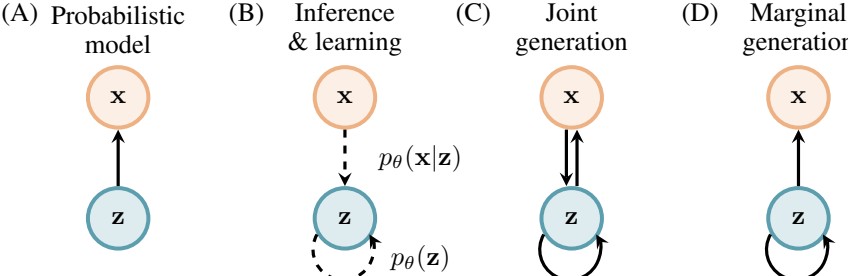

Figure 1: (A) The directed graphical model of the energy-based model (EBM). (B) The latent variable $\mathbf{z}$ receives the likelihood information $p_\theta(\mathbf{x}|\mathbf{z})$ from the observation $\mathbf{x}$ and combines it with the prior knowledge $p_\theta(\mathbf{z})$ to perform the inference dynamic. And the connected weights $\theta$ changes following the learning dynamic (dashed line). (C) The latent variable $\mathbf{z}$ performs inference dynamic and the observation $\mathbf{x}$ performs generation dynamic, which leads to the distribution of $\mathbf{x}, \mathbf{z} \sim p_\theta(\mathbf{x}, \mathbf{z})$. And the connection weights $\theta$ are fixed (solid line). (D) The latent variable doesn't receive likelihood information from observation and $\mathbf{z} \sim p_\theta(\mathbf{z})$, $\mathbf{x} \sim p_\theta(\mathbf{x})$ which is called marginal generation.

The neural system can adopts the gradient based learning method such as gradient decent. And the gradients is calculated as (see SI for detailed proof),

$$\nabla_\theta D_{\mathrm{KL}}\left[p_{\mathrm{true}}(\mathbf{x}) \parallel p_\theta(\mathbf{x})\right] = -\mathbb{E}_{\mathbf{x} \sim p_{\mathrm{true}}(\mathbf{x})} \mathbb{E}_{\mathbf{z} \sim p_\theta(\mathbf{z}|\mathbf{x})}\left[\nabla_\theta \ln p_\theta(\mathbf{x}, \mathbf{z})\right]. \tag{2}$$

The second expectation of the above equation requires the posterior of a given observation $\mathbf{x}$, which is calculated as $p_\theta(\mathbf{z}|\mathbf{x}) = p_\theta(\mathbf{x}, \mathbf{z})/p_\theta(\mathbf{x})$. Practically, the posterior is intractable for the denominator $p_\theta(\mathbf{x})$ requires complex integral. Variational inference is usually used to solve this problem (such as VAE [33]). And the EBMs adopts the sampling method to avoid complex calculations. By leveraging the relationship $\nabla_{\mathbf{z}} \ln p_\theta(\mathbf{z}|\mathbf{x}) = \nabla_{\mathbf{z}} \ln p_\theta(\mathbf{x}, \mathbf{z})$, the samples of posterior can be offered by the neural dynamic (Langevin sampling),

$$\tau_z \frac{\mathrm{d}\mathbf{z}}{\mathrm{d}t} = \nabla_{\mathbf{z}} \ln p_\theta(\mathbf{x}, \mathbf{z}) + \sqrt{2\tau_z}\boldsymbol{\xi}, \tag{3}$$

where $\xi$ is Gaussian white noise and $\tau_z$ is the time constant. The stationary distribution of the above dynamic is our target distribution $p_\theta(\mathbf{z}|\mathbf{x})$. The sampling algorithm, along with the joint distribution, determines the connections of neurons and their inference dynamic (Fig.1B).

For the learning dynamic, the brain receives the observations from the real world continuously ($\mathbb{E}_{\mathbf{x} \sim p_{\mathrm{true}}(\mathbf{x})}$), and the neural dynamic mentioned above can produce samples of their posterior simultaneously ($\mathbb{E}_{\mathbf{z} \sim p_\theta(\mathbf{z}|\mathbf{x})}$). The parameters can be updated according to the gradient, which is often implemented by Hebbian learning rules(Fig.1B),

$$\tau_\theta \frac{\mathrm{d}\theta}{\mathrm{d}t} = -\nabla_\theta D_{\mathrm{KL}}\left[p_{\mathrm{true}}(\mathbf{x}) \parallel p_\theta(\mathbf{x})\right] = \nabla_\theta \ln p_\theta(\mathbf{x}, \mathbf{z}), \tag{4}$$

where $\tau_\theta$ is the time constant of synapses.

For neuroscience, this is the end of the story. Nevertheless, EBMs can also generate observations which the machine learning society follow with interest. In the generation process, the observation is not fixed but undergoes the Langevin sampling,

$$\tau_x \frac{\mathrm{d}\mathbf{x}}{\mathrm{d}t} = \nabla_{\mathbf{x}} \ln p_\theta(\mathbf{x}, \mathbf{z}) + \sqrt{2\tau_x}\boldsymbol{\xi}. \tag{5}$$

The latent variable $\mathbf{z}$ can follow the inference dynamic Eq.(3). In this case, the observation $\mathbf{x}$ and latent variable $\mathbf{z}$ together follow the joint distribution $p_\theta(\mathbf{x}, \mathbf{z})$. Thus, the observation $\mathbf{x}$ can produce samples following $p_\theta(\mathbf{x})$, which is called joint generation (Fig.1C). However, in order to get the marginal distribution $p_\theta(\mathbf{x})$, the latent variable $\mathbf{z}$ just needs to follow the prior distribution $p_\theta(\mathbf{z})$ rather than reaching the posterior. In this case, the generation dynamic of latent variable is written as,

$$\tau_z \frac{\mathrm{d}\mathbf{z}}{\mathrm{d}t} = \nabla_{\mathbf{z}} \ln p_\theta(\mathbf{z}) + \sqrt{2\tau_z}\boldsymbol{\xi}. \tag{6}$$

And the stationary distribution of Eq.(5) performed by observation $\mathbf{x}$ still equals to $p_\theta(\mathbf{x})$, which is called marginal generation (Fig.1D). In Sec.5, we will see that the marginal generation performs better than joint generation. Here is an informal understanding. The process of sampling $(\mathbf{x}, \mathbf{z})$ can be understood as searching for a specific pair. We assume that the size of the $\mathbf{x}$-space is $O(m)$ and the size of the $\mathbf{z}$-space is $O(n)$. In joint generation, the search is conducted simultaneously in both the $\mathbf{x}$-space and $\mathbf{z}$-space, resulting in a required search space of $O(n * m)$. In marginal generation, the process involves initially searching in the $\mathbf{z}$-space according to $p_\theta(\mathbf{z})$. Once $\mathbf{z}$ is found, it is fixed. This step's search space size is $O(n)$. Then, $\mathbf{x}$ is searched based on $p_\theta(\mathbf{x}|\mathbf{z})$ in the $\mathbf{x}$-space. This step's search space size is $O(m)$, leading to a combined required search space size of $O(n + m)$.

## 3 Exponential-family energy-based model

In this section, we provide a neural implementation of the HEE model and outline the specific dynamics involved in inference, learning, and generation, as discussed in Section 2. Approximating the target distribution $p_{\text{true}}(\mathbf{x})$ which is diverse and complex requires a good representation ability of the model. Exponential families include many of the most common distributions (such as normal, Poisson, gamma distribution and so on). Moreover, exponential families can be easily parameterized, allowing for generalization and flexibility in modeling various types of distributions.

Let $\mathbf{x}_0 \in \mathbb{R}^{n_0}$ be the observation (such as an image) received by our brain. And there are $L$ layers of neurons representing the latent variables $\mathbf{x}_{1:L} = \{\mathbf{x}_1, \mathbf{x}_2, ..., \mathbf{x}_L\}$, $\mathbf{x}_l \in \mathbb{R}^{n_l}$. The joint distribution is a Markov chain (Fig.2A) starting with $p(\mathbf{x}_L) = \exp\left[\boldsymbol{\eta}_L^T \phi(\mathbf{x}_L) + g(\mathbf{x}_L) - A(\boldsymbol{\eta}_L)\right]$,

$$p_\theta(\mathbf{x}_{0:L}) = p(\mathbf{x}_l) \prod_{l=0}^{L-1} p_\theta(\mathbf{x}_l|\mathbf{x}_{l+1}), \quad p_\theta(\mathbf{x}_l|\mathbf{x}_{l+1}) = \exp\left[\boldsymbol{\eta}_l^T \phi(\mathbf{x}_l) + g(\mathbf{x}_l) - A(\boldsymbol{\eta}_l)\right]. \quad (7)$$

The natural parameter $\boldsymbol{\eta}_l \in \mathbb{R}^{n_l}$ is a function of $\mathbf{x}_{l+1}$ with parameters $\theta_l \in \mathbb{R}^{n_l \times n_{l+1}}$, which is written as $\boldsymbol{\eta}_l = \theta_l f(\mathbf{x}_{l+1})$, where $f(\cdot)$ is the activation function. And $\boldsymbol{\eta}_L$ is constant. The sufficient statistic $\phi(\mathbf{x}_l) \in \mathbb{R}^{n_l}$ and the base measure $g(\mathbf{x}_l) \in \mathbb{R}$ is the function of $\mathbf{x}_l$. $A(\boldsymbol{\eta}_l) \in \mathbb{R}$ is the normalize term (log-partition function) to make sure the sum of the probability equals to 1. In order to get the inference dynamic, we substitute the joint distribution Eq.(7) into the Langevin dynamic Eq.(3) obtaining,

$$\tau_z \frac{\mathrm{d}\mathbf{x}_l}{\mathrm{d}t} = f'(\mathbf{x}_l)\theta_{l-1}^T \left[\phi(\mathbf{x}_{l-1}) - A'(\boldsymbol{\eta}_{l-1})\right] + \phi'(\mathbf{x}_l)\boldsymbol{\eta}_l + g'(\mathbf{x}_l) + \sqrt{2\tau_z}\xi_l, \quad (8)$$

where $f'(\mathbf{x}_l), \phi'(\mathbf{x}_l) \in \mathbb{R}^{n_l \times n_l}$ are diagonal matrices. The derivative of log-partition $A'(\boldsymbol{\eta}_{l-1})$ is intractable for it needs complex integral. Here, we use a group of interneurons $\boldsymbol{\varepsilon}_{l-1} \in \mathbb{R}^{n_{l-1}}$ to represent the term $\phi(\mathbf{x}_{l-1}) - A'(\boldsymbol{\eta}_{l-1})$. It can be proved that $A'(\boldsymbol{\eta}_{l-1}) = E_{\mathbf{x}_{l-1} \sim p_\theta(\mathbf{x}_{l-1}|\mathbf{x}_l)}\left[\phi(\mathbf{x}_{l-1})\right]$ (See SI for detailed proof). Thus, in order to calculate $A'(\boldsymbol{\eta}_{l-1})$, the interneurons need to produce samples $\mathbf{x}_{l-1}$ following the distribution $p_\theta(\mathbf{x}_{l-1}|\mathbf{x}_l)$ in a short time compared with the inference dynamic. Therefore, the dynamic of $\boldsymbol{\varepsilon}_{l-1}$ can be written as,

$$\boldsymbol{\varepsilon}_{l-1} = \phi(\mathbf{x}_{l-1}) - \phi(\mathbf{u}_{l-1}), \quad \tau_u \frac{\mathrm{d}\mathbf{u}_{l-1}}{\mathrm{d}t} = \phi'(\mathbf{u}_{l-1})\boldsymbol{\eta}_{l-1} + g'(\mathbf{u}_{l-1}) + \sqrt{2\tau_u}\xi_u. \quad (9)$$

By setting the time constant $\tau_u \ll \tau_z$ [1], we can ensure that $\mathbf{u}_{l-1}$ converges much faster than $\mathbf{x}_l$, and the stationary distribution of $\mathbf{u}_{l-1}$ corresponds to $p_\theta(\mathbf{u}_{l-1}|\mathbf{x}_l)$. This leads to $\boldsymbol{\varepsilon}_{l-1} = \phi(\mathbf{x}_{l-1}) - A'(\boldsymbol{\eta}_{l-1})$.

Now, we can rewrite the inference dynamic Eq.(8) into,

$$\tau_z \frac{\mathrm{d}\mathbf{x}_l}{\mathrm{d}t} = f'(\mathbf{x}_l)\theta_{l-1}^T \boldsymbol{\varepsilon}_{l-1} + \phi'(\mathbf{x}_l)\boldsymbol{\eta}_l + g'(\mathbf{x}_l) + \sqrt{2\tau_z}\xi_l. \quad (10)$$

The first term on the right side of the dynamic equation indicates that neurons $\mathbf{x}_l$ receive feedback from interneurons $\boldsymbol{\varepsilon}_{l-1}$, which provides likelihood information $p_\theta(\mathbf{x}_{l-1}|\mathbf{x}_l)$. The second term shows

---

[1]There are various types of interneurons that target on pyramidal cells, comprising approximately 10-20% of the overall neuron population in the cerebral cortex [34]. The interneurons in the HEE model bear the closest resemblance to the Large Basket Cell or Nest Basket Cell [35], which collectively constitute around 50% of interneurons. Their electrophysiological characteristics include fast spiking, non-accommodating, and non-adapting behaviors. These interneurons have also been identified in the visual cortex of ferrets [36], displaying short-duration action potentials (approximately 0.5 ms at half height). This suggests that these neurons have shorter time constants compared to pyramidal cells.

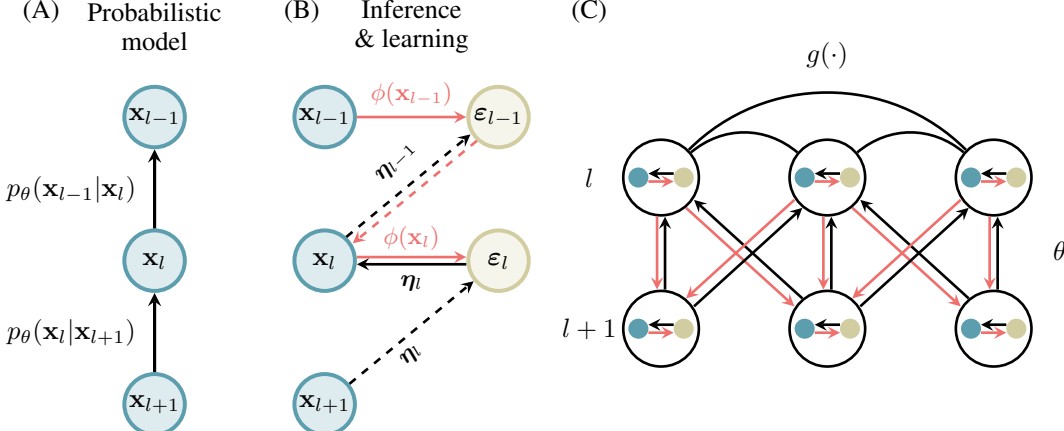

Figure 2: (A) The directed graphical model of the hierarchical exponential-family energy-based (HEE) model . (B) The inference and learning dynamic of HEE model. The red arrows represent the likelihood information and the black arrows represent the prior information. Neurons $\mathbf{x}_l$ receive the likelihood information from $\boldsymbol{\varepsilon}_{l-1}$ and receive prior information $\boldsymbol{\eta}_l$ from $\boldsymbol{\varepsilon}_l$. The interneurons $\boldsymbol{\varepsilon}_{l-1}$ receive the natural parameter $\boldsymbol{\eta}_{l-1}$ from neurons $\mathbf{x}_l$ and perform the Langevin sampling dynamic to approximate $A'(\boldsymbol{\eta}_{l-1})$. Then, the interneurons compare it with the the sufficient statistic $\phi(\mathbf{x}_{l-1})$ received from $\mathbf{x}_{l-1}$ to calculate the value of $\boldsymbol{\varepsilon}_{l-1}$. The dashed line shows that the connection weights $\theta$ will perform gradient decent in the learning dynamic. (C) The neural connection diagram between layer $l$ and layer $l + 1$.

that neurons $\mathbf{x}_l$ receive prior knowledge $p_\theta(\mathbf{x}_l|\mathbf{x}_{l+1})$ from interneurons $\boldsymbol{\varepsilon}_l$ through a feedforward loop. The term $g'(\mathbf{x}_l)$ controls the self-connections within layer $l$ (Fig.2B). $\boldsymbol{\varepsilon}_{l-1}$ is also called the error term in PCNs. Here, we show that the predictions in PCNs essentially estimate the log-partition.

Then, we can obtain the learning dynamic by substituting the joint distribution Eq.(7) into the gradient decent dynamic Eq.(4),

$$\tau_\theta \frac{\mathrm{d}\theta_l}{\mathrm{d}t} = [\phi(\mathbf{x}_l) - A'(\boldsymbol{\eta}_l)] f(\mathbf{x}_{l+1})^T = \boldsymbol{\varepsilon}_l f(\mathbf{x}_{l+1})^T. \tag{11}$$

The derivatives of the log-partition function are stored in the interneurons $\boldsymbol{\varepsilon}_l$. As a result, the synaptic changes are determined solely by local neurons, adhering to Hebbian rules.

After inference and learning, our model can also generate observations. During joint generation, the dynamics of neurons $\mathbf{x}_{1:L}$ follow the same principles as the inference dynamics described by Eq.(10). By substituting the joint distribution Eq.(7) into the Langevin dynamic Eq.(5), we can generate new samples from the model by,

$$\tau_x \frac{\mathrm{d}\mathbf{x}_0}{\mathrm{d}t} = \phi'(\mathbf{x}_0)\boldsymbol{\eta}_0 + g'(\mathbf{x}_0) + \sqrt{2\tau_x}\xi_0 \tag{12}$$

And for marginal generation, we can substitute the prior distribution described in Eq.(7) into Eq.(6) obtaining the dynamic of neurons $\mathbf{x}_{1:L}$,

$$\tau_z \frac{\mathrm{d}\mathbf{x}_l}{\mathrm{d}t} = \phi'(\mathbf{x}_l)\boldsymbol{\eta}_l + g'(\mathbf{x}_l) + \sqrt{2\tau_z}\xi_l. \tag{13}$$

## 4 Neural adaptation accelerate the sampling process

Langevin sampling essentially performs random walks in local regions rather than the whole posterior space [37], because the drift term $\nabla_\mathbf{z} \ln p_\theta(\mathbf{x}, \mathbf{z})$ in Eq.(3) will vanish near the local minima and only noise term remains (Fig.3A). Sampling the entire posterior space is a time-consuming process and does not align with the brain's ability to perform tasks quickly. Therefore, it is important to implement a faster sampling algorithms for our model.

In this section, we show that by including noisy adaptation, the network is able to speed up the inference dynamic significantly. Adaptation is a common phenomenon observed in neural systems,

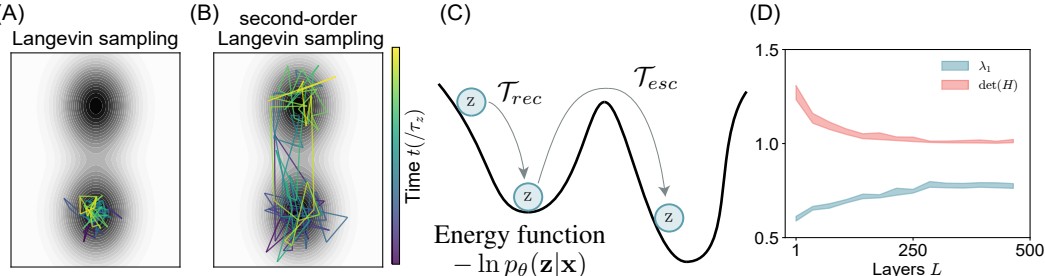

Figure 3: (A)(B) The sampling trace of Langevin dynamic and second-order Langevin dynamic of a mixture of 2D Gaussian distribution. (C) Illustration of the sampling process. States with lower energy indicates a higher probability, which needs to be stay longer. And the network needs to cross the energy barrier to reach a new local minima for the non-convex of the energy function. (D) $\lambda_1$ and $\det(H)$ increases and decreases, respectively, with the changes of layers $L$ and the total number of the neurons fixed ($\sum_l n_l = 10000$).

where negative feedback mechanisms are employed to suppress neuronal responses when they reach high levels. Here, we show that the neural adaptation $\mathbf{v}$ can introduce an auxiliary variable to implement a faster sampling algorithm, which is called second-order Langevin dynamics (SLD),

$$\tau_z \frac{d\mathbf{z}}{dt} = \nabla_{\mathbf{z}} \ln p_\theta(\mathbf{z}|\mathbf{x}) - \mathbf{v} + \sqrt{2\tau_z}\xi, \tag{14}$$

$$\tau_v \frac{d\mathbf{v}}{dt} = -\frac{m\mathbf{v}}{2} + m\sqrt{2\tau_v}\xi, \tag{15}$$

where $m > 0$ controls the adaptation strength. The auxiliary variable $\mathbf{v}$ can keep the network moving while reaching the local minima, which essentially serves as the momentum term (Fig.3B). When there is no adaptation ($m = 0$), the above dynamic will degenerate to the Langevin sampling (LS) described in Eq.(3). And the stationary distribution of the above dynamic remains to be $p_\theta(\mathbf{z}|\mathbf{x})$ (See SI for proof). By substituting the joint distribution of exponential family Eq.(7) into the above dynamic, we can obtain the SLD inference dynamic,

$$\tau_z \frac{d\mathbf{x}_l}{dt} = f'(\mathbf{x}_l)\theta_{l-1}^T \boldsymbol{\varepsilon}_{l-1} + \phi'(\mathbf{x}_l)\boldsymbol{\eta}_l + g'(\mathbf{x}_l) - \mathbf{v}_l + \sqrt{2\tau_z}\xi_l, \tag{16}$$

$$\tau_v \frac{d\mathbf{v}_l}{dt} = -\frac{m\mathbf{v}_l}{2} + m\sqrt{2\tau_v}\xi_l. \tag{17}$$

Here we exemplify the neural adaptation with spike frequency adaptation (SFA) [38]. In the neural system, the adaptation current $\mathbf{v}_l$ accumulates the noise term $\xi_l$ coming from the ion concentrations, release of neural transmitters, activation/inactivation of ion channels and so on, which is described by Eq.(17). The adaptation current induces suppression on neurons $\mathbf{x}_l$, acting as momentum variables to accelerate the sampling process (Eq.(16)).

We conduct further investigation to elucidate the precise mechanism by which noisy adaptation facilitates the acceleration of the sampling process in the HEE model. Considering that the energy function $-\ln p_\theta(\mathbf{x}_{1:L}|\mathbf{x})$ is non-convex, the sampling process can be divided into two parts. In the first part, the network needs to find a local minima and samples near the local minima, which takes a certain amount of time called recurrence time $\mathcal{T}_{rec}$ (Fig.3C). In the second part, the network needs to leave the local minima and find a new one, which takes a certain amount of time called the escape time $\mathcal{T}_{esc}$. Typically, we have $\mathcal{T}_{rec} \ll \mathcal{T}_{esc}$. The total time to get stationary distribution can be approximated by $\mathcal{T} = \mathcal{T}_{esc} + \mathcal{T}_{rec}$. It can be proved that [39] the recurrence time is bounded by $\mathcal{T}_{rec} = \mathcal{O}\left(1/\lambda_1(H_J)\right)$. $\lambda_1(H_J)$ is the smallest eigenvalue of the matrix $H_J$,

$$H_J = \begin{pmatrix} H/\tau_z & I/\tau_z \\ 0 & mI/(2\tau_v) \end{pmatrix} \tag{18}$$

where $H$ is the Hessien matrix of the energy function $-\ln p_\theta(\mathbf{x}_{1:L}|\mathbf{x}_0)$. The smallest eigenvalue of $H_J$ is calculated as $\lambda_1(H_J) = \min\{\lambda_1(H)/\tau_z, m/(2\tau_v)\}$. Thus, in the case $m > 2\lambda_1(H)\tau_v/\tau_z$, the SLD can reduce the recurrence time $\mathcal{T}_{rec}$ to accelerate the sampling process. And the escape

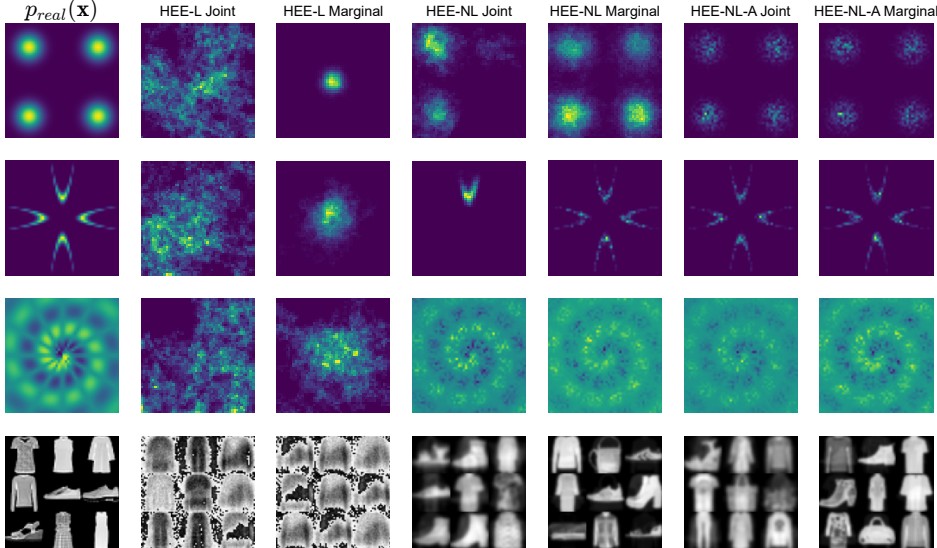

Figure 4: Evaluation on *2D synthetic datasets* and *FashionMNIST*: a mixture of four Gaussian distribution (first line), a mixture of four banana-shaped distribution (second line), pinwheel-shaped distribution (third line).

time is bounded by $\mathcal{T}_{esc} = \mathcal{O}\left(\sqrt{1/\det(H_J)}\right)$. The determinant of $H_J$ is calculated as $\det(H_J) = \det(H)(m\tau/2\tau_v)^{\sum n_l}$. Thus, $\det(H_J)$ is monotonically increaseing with $m$, indicating that the larger $m$ is, the shorter the time it takes to escape from the local minima.

The analysis presented above demonstrates that the convergence speed of the inference dynamic is determined by the values of $\lambda_1(H)$ and $\det(H)$. This valuable insight can be leveraged to guide the design of the network architecture, enabling the creation of more efficient and effective models. Specifically, with a fixed total number of neurons $\sum_l n_l$, increasing the number of layers $L$ results in a deeper network, while decreasing the number of layers results in a wider network. Practically, we show that $\det(H)$ will decrease with layers $L$ while $\lambda_1$ will increase (Fig.3D), which indicates that there is a trade-off between the recurrence time $\mathcal{T}_{rec}$ and the escaping time $\mathcal{T}_{esc}$ with different layers $L$ (See SI for detailed setting and analysis).

## 5 Experiment

In this section, we firstly validate the capability of HEE model for approaching complex distribution by examining the quality of generation. Then, we show that our model demonstrates similarity in the representation of natural images to the biological visual system. And adaptation can induce oscillatory behavior and transient overshoots in neurons during the inference phase.

### 5.1 Generation

Firstly, we conducted experiments using three variations of the HEE model, each with different $\phi(x)$ functions and sampling methods (Tabel 1), to evaluate their capabilities. The experiments were performed on both *2D synthetic datasets* and the *FashionMINST*. We use the fully connected architecture, i.e., $\theta_l$ has no zero elements. The results (second and third column) show that HEE with linear statistic (HEE-L) struggles to capture the complex distribution. Moreover, we theoretically prove that HEE-L can only approach unimodal distributions (See SI for detailed proof). For HEE-NL, some modes are missing while using the joint generation. And when the spacing between modes is large, there is an issue of non-uniformity among different modes. And the marginal generation converge much faster than the joint generation. In *FashionMNIST*, it takes less time for marginal method to get the generation of high-quality images.

Then, we employ the HEE-NL-A with layers $L = 10$ on the *CIFAR10* unconditional. The sparse connection is employed as a method to mimic the receptive field behavior found in biological systems. We quantitatively evaluate image quality of HEE-NL-A with Inception score [40] and FID score [41] in

| Model | $\phi(x)$ | Sampling method |
|---|---|---|
| HEE-L | $x$ | LS |
| HEE-NL | $sigmoid(x)$ | LS |
| HEE-NL-A | $sigmoid(x)$ | SLD |

Table 1: Table of different HEE models.

Tabel 2. Overall, we achieve a performance comparable with the previous EBMs. And the generation quality of the marginal method is better than joint method, which agrees with the previous results [42].

## 5.2  Inference

We further use the HEE-NL-A trained on *CIFAR10* to explore the relationship between the latent features and semantic information, including orientation, color and category.

*Orientation:* Simple cells [43] and complex cells [44] are the most prominent and widely observed neurons in the biological visual system that exhibit tuning to orientation. They are found in the primary visual cortex (V1) of numerous animal species [45, 46]. We present the model with gabor images of different orientations commonly used in experiments (Fig.6A) and compute the mean and variance of the neural responses for each neuron. Then, we use a Gaussian curve with bandwidth limited from 20° to 90° and a two-modes Gaussian curve to fit the simple cell and complex cell, respectively (Fig.6A&B). We find that the proportion of simple cells and complex cells remains relatively consistent across each layer and both decrease with increasing layers in our model (Fig.6C).

*End stopping:* In the HEE model, interneurons essentially represent the error term in the PCNs. We have observed the phenomenon of 'end stopping' in interneurons (Fig.6D), which aligns with the end-stopping behavior observed in error neurons in the PCNs [21].

*Color:* Recent study shows that there is a hierarchical representation for chromatic processing across the ventral pathway of macaque [47]. We present our model using reshaped natural images [48] and employ principal component analysis to demonstrate that the middle layer's neural representation's most informative dimensions carry chromatic information (Fig.6E).

*Category:* Visual object recognition is believed to be solved by the brain hierarchically [49]. A recent study [50] demonstrate that the inferotemporal cortex, situated in the deeper layer of the visual pathway, is capable of constructing a linear map of the object space. For each layer in our model, we employed a linear support vector machine (SVM) to classify the ten labels of the *CIFAR10*. The SVM was trained using the average neural responses as features. Fig.6F illustrates the projection of the features in the last layer onto the SVM weights corresponding to the "cat" and "dog" labels. Furthermore, we observe that the classification accuracy improves as we move up the layers of our model (Fig.6G), which is consistent with findings in both biological visual systems [50] and the artificial neural networks [51].

*Phenomena:* Oscillations [52] and transients [53] are two kinds of spatial-temporal dynamic features in neural systems, which play a crucial role during the sampling process [13]. Here, we show that by adjusting the adaptation strength $m$, the oscillation frequency of the HEE model can span within the

| Model | IS | FID |
|---|---|---|
| HEE-NL-A (Joint) | 5.95 | 43.21 |
| EBM (single) [27] | 6.02 | 40.58 |
| HEE-NL-A (Marginal) | 6.47 | 37.05 |
| MEG (Generator) [42] | 6.49 | 35.02 |
| EBM (10 ensemble) | 6.78 | 38.20 |
| MEG (MCMC) | 7.31 | 33.18 |

Table 2: Table of Inception and FID scores.

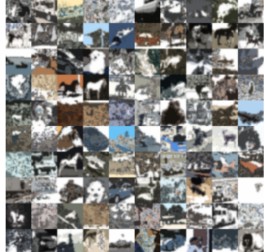

Figure 5: Marginal generation of HEE-NL-A on CIFAR10.

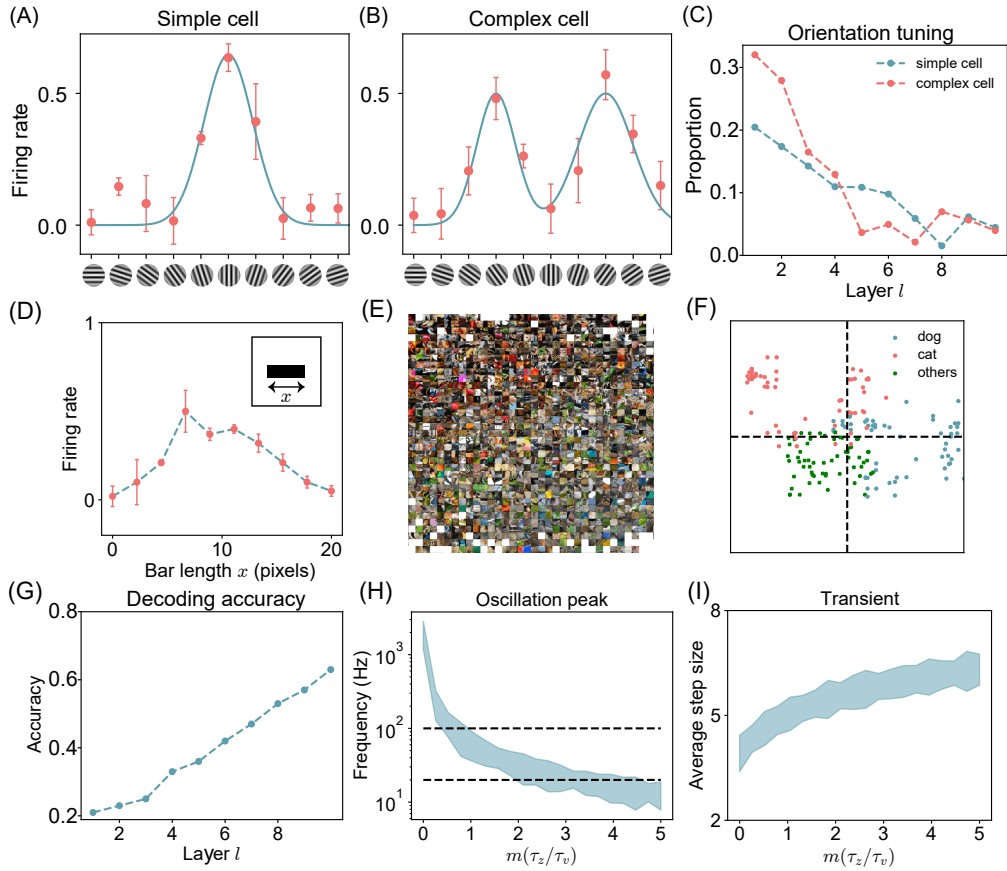

Figure 6: (A)(B) The red dots show the average firing rate of two neurons in $\mathbf{x}_1$ with different gabor-like stimulus. Different tuning curves are used to fit simple cells and complex cells. (C) The proportion of simple cells and complex cells in each layer. (D) We show horizontal bars of varying lengths to the HEE. The red dots show the average firing rate of a neuron in $\boldsymbol{\epsilon}_1$ whose corresponding neuron in $\mathbf{x}_1$ is a simple cell preferring 0 degree. (E) The images are plotted at the location corresponding to the projection of their average neural response in layer 5 onto the first two principal components. Red images are located in the upper left corner, while blue-green images are located in the lower right corner. (F) The true label of cat, dog, and other categories should be respectively located in the second, fourth, and third quadrants. (G) The classification accuracy of the SVM in each layer. (H) We sampled and statistically analyzed the distribution of the highest firing rate frequencies of all neurons during the inference phase in the first $100\tau_z$ for different values of $m$. The gamma band is centered around the dashed line. (I) We sampled and statistically assessed the maximum change in firing rates of all neurons during the inference phase in the first $100\tau_z$ for different values of $m$. We refer to the mean of the maximum change values as the 'average step size'. We utilize the average step size of the neurons during the sampling process as an indicator of transients.

range of 20-80 Hz (gamma band), which is widely observed in visual systems [54] (Fig.6H). And stimulus-onset transients of the firing rate can also be enhanced by the adaptation (Fig.6I).

## 6  Discussion

The present study investigates the sampling-based inference and learning dynamic within the framework of an intrinsic generative model. We introduce the HEE model as a neural implementation that utilizes neural dynamics and Hebbian learning. Additionally, we demonstrate that the inclusion of

neural adaptation can significantly accelerate the sampling process and give rise to various dynamic phenomena throughout the network. In this section, we will discuss several related theories and models.

**Probabilistic Population Code (PPC)** [55, 56] is another theory that explains how the brain perform Bayesian inference, in which neural responses are interpreted as the parameters of the probability distributions. We adopted the idea [57] that PPC theory incorporates two generative models. In the framework of PPC, the experimenter presents the subject with observation $\mathbf{x}$ (gabor image) based on the semantic information $\mathbf{s}$ (orientation), which actually defines an external generative model from $\mathbf{s}$ to $\mathbf{x}$. And the subjects holds an intrinsic generative model with latent variable $\mathbf{z}$ represented by neural response to interpret the observation $\mathbf{x}$. PPC theory integrates two generative models into a single generative model, in which the neural response $\mathbf{z}$ is regarded as the observation generated from the semantic information $\mathbf{s}$. We propose that the learning dynamic occurs exclusively within the intrinsic generative model, as the brain is not aware of the external generative model.

**Energy-based models (EBMs)** [20, 58] When EBMs were initially proposed [20], they had latent variables corresponding to neurons. Later, to enhance the model's expressive power, hierarchical structures were introduced [59, 60]. Such EBMs could typically ensure local learning in space. As artificial neural networks have become increasingly powerful, it has been observed that for generative tasks, there's no need to explicitly introduce neurons as latent variables within EBMs. Instead, one can directly employ a neural network to represent the energy [27]. Training such EBMs often involves utilizing BP. The distinction between these EBMs and traditional EBMs is akin to the difference between dynamic systems and recurrent neural networks.

Regardless of whether it's the traditional EBMs or the new type of EBMs, both involve the challenge of estimating the partition function. This difficulty arises from the fact that as the depth of the energy function increases, the total sample space required multiplies the space for each layer. In the case of HEE, we allocate the partition function across each layer. As a result, the total sample space required is the sum of the spaces for each layer. This significantly reduces the required sample space.

**Predictive coding networks (PCNs)** [21, 22] The interneurons in the HEE model serve a similar role to the prediction error in PCNs. And our theoretical analysis shows that the predictions in PCNs essentially represent the decomposed log-partition function. And PCNs don't stress the sampling-based inference, which requires them to approximate the energy function using variation inference by delta function. Sampling-based inference can assist the network in exploring the posterior probability space, leading to a more accurate estimation of the energy function. Additionally, it can account for the observed neural variability in experiments.

**Diffusion models (DDPMs)** [61, 62] The HEE model and DDPMs share the same joint distribution and both exhibit a hierarchical Markov structure, which may contribute to the HEE model's strong expressive potential. The marginal generation is also called latent space MCMC [63, 42], which is similar to the generation process of DDPMs. While DDPMs unfolds the Markov chain over time, the HEE model unfold it between layers of neurons. However, in order to reach a better performance, DDPMs use a fixed diffusion process as the inference dynamic, which may not be adopted by our brain since the latent variables in our brain carry semantic information (such as simple cells [43]).

**Speed up sampling** [17, 38] In previous work, inhibition neurons were used to serve as momentum terms to accelerate sampling [17]. However, this approach required a one-to-one correspondence between inhibition neurons and excitatory neurons. In our approach, we consider the adaptive properties inherent in each neuron itself to serve as momentum, naturally resolving the one-to-one correspondence issue. Furthermore, our consideration extends to sampling in a non-convex energy space, which differs from the prior focus solely on convex space convergence properties [38].

## Acknowledgement

I'd like to express my gratitude to Tianqiu Zhang and Chaoming Wang for their assistance in configuring the experimental environment. I also want to thank BrainPy [64] for their support throughout this work. Special thanks to Yumeng Cao for reviewing the article's grammar and expression. This work was supported by Science and Technology Innovation 2030-Brain Science and Brain-inspired Intelligence Project (No. 2021ZD0200204).

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
