# OpenReview forum: "Neural Sampling in Hierarchical Exponential-family Energy-based Models"
_NeurIPS.cc/2023/Conference — NeurIPS 2023 poster_

### Official Review · Reviewer_moHy · 2023-06-24

**Soundness:** 4 excellent
**Presentation:** 4 excellent
**Contribution:** 3 good
**Rating:** 8
**Confidence:** 3

**Summary:**

The paper introduces the hierarchical exponential family energy based model as a biologically plausible mechanism for the brain to interpret the external world. The authors first describe the learning/inference and the generation process of the model, the dynamics is local and biologically plausible through introducing a set of fast interneurons for the log-partition functions. They further show that adding adaptation mechanism can accelerate the sampling process. Finally, the authors present extensive numerical results to show that the model achieves good generation quality, and exhibit representations similar to those in the biological visual systems.

**Strengths:**

1. This paper presents a nice combination of theoretical analysis and extensive numerical results to demonstrate that their model exhibits multiple desired properties (including the acceleration effect of the adaptation mechanism, generation quality and representations).
2. The authors show multiple interesting aspects of their models that are very relevant to the neuroscience community, including the acceleration effect of neural adaptation, and the similarity between the representations in biological visual system and the model, thereby providing new insights on the possible advantages/origins of well-known neural mechanisms/representations in the context of Bayesian inference in the brain.


**Weaknesses:**

The paper is very well written in general, and establishes nice connections between their findings and neural representations in the visual system. However, the comparison with neuroscience is limited to well-known experimental findings, it would be nice if the authors can add a discussion to highlight several possible experimental predictions of their model.

**Questions:**

1. Eq4: It is a little confusing, before the second equality there is the KL divergence which is defined by Eq.2 with averaging over $z$ and $x$, whereas after the equality it is actually $p_{\theta}(x,z)$ for a sample of $\{x,z\}$. Maybe it would be better to introduce separate notations for the random variables $x$ and $z$ (perhaps capital letters) and the realizations of $x$ and $z$.
2. Line 99-100: If I understand correctly, both joint generation and marginal generation can generate samples following  $p_{\theta}(x)$? Why say “However, in order to get the marginal distribution $p_{\theta}(x)$” here?
3. What could the fast interneurons $\epsilon_l$ correspond to in the visual circuitry?
4. Is there an intuitive explanation why the marginal method always seem to perform better than the joint method for generation?
5. Fig 6D: As the proportion of both simple and complex cells decrease with the number of layers, do you see neurons with more complicated receptive fields (objects, etc.)?


**Limitations:**

Yes the authors have addressed the limitations.

---

> ### Author Rebuttal · Authors · 2023-08-09
>
> We acknowledge the encouraging and valuable comments of the reviewer, and would like to address the concerns of the reviewer in details below.
>
> **Weaknesses:**
>
> We are grateful for the issues the reviewer highlighted. In the present study, as the first step, we aim to establish a framework, which addresses a particularly challenging problem in theoretical neuroscience, i.e., achieving simultaneous integration of biologically plausible local learning and sampling-based inference within a hierarchical network structure. In our forthcoming work, we will explore in details how model predictions can be experimentally validated. Recently, we have also developed a keen interest in inputs with temporal information and are conducting relevant experiments on stimuli with serial dependence [1]. We will incorporate some discussions about the latest developments in the revised manuscript.
>
> [1] Fischer, Jason, and David Whitney. "Serial dependence in visual perception." Nature neuroscience 17.5 (2014): 738-743.
>
> **Questions:**
> 1. Thanks for the suggestion. In the revised version, we will distinguish between random variables and their samples. For example, Eq. (2) will be revised as follows: $\nabla D_{KL}\left[p_{true}(\mathbf{x})\parallel p_{\theta}(\mathbf{x})\right]=-E_{\tilde{\mathbf{x}}\sim p_{true}(\mathbf{x})}E_{\tilde{\mathbf{z}}\sim p_\theta(\mathbf{z}|\tilde{\mathbf{x}})}\left[\nabla_\theta\ln p_\theta({\tilde{\mathbf{x}},\tilde{\mathbf{z}}})\right]$, where $\mathbf{x},\mathbf{z}$ are random vairables and  $\tilde{\mathbf{x}}, \tilde{\mathbf{z}}$ are samples
>
> 2. Indeed, both joint generation and marginal generation can produce samples following $p_{\theta}(\mathbf{x})$. We recognize that our expression was unclear. To clarify, joint generation involves simultaneous sampling in the x-space and z-space, leading to x's marginal distribution following $p_{\theta}(\mathbf{x})$; whereas, marginal generation first samples in the z-space and then generates x according to $p_\theta(\mathbf{x}|\mathbf{z})$, which also results in the x's marginal distribution conforming to $p_{\theta}(\mathbf{x})$. We will clarify this in the revised manuscript.
>
> 3. Please see **rebuttal to all reviewers (part 2)**.
>
> 4. Joint generation involves simultaneous sampling in both x-space and z-space according to $p_\theta(x,z)$. On the other hand, marginal generation first samples from z-space according to $p_\theta(z)$, and after obtaining $z$ samples, proceeds to sample from x-space following $p_\theta(x|z)$. Considering the time complexities of $O(m)$ for x-space and $O(n)$ for z-space sampling, joint generation has a time complexity of $O(m*n)$, while marginal generation's complexity is $O(m+n)$. This implies that marginal generation is more likely to yield high-quality samples as its sampling space is smaller.
>
> 5. We have investigated not only neurons of orientation tuning, but also neurons tuned to high-order features [1] and neurons responsive to image categories. However, the proportion of these neurons was extremely low, constituting less than 1%. Therefore, we chose not to report these results.
>
>     Although only a small number of neurons exhibit selectivity for categories at the single-neuron level, information about categories can still be linearly decoded from the population response. We used neural representations in each layer to train a linear SVM to discriminate the categories of CIFAR-10 images. We found that the classification accuracy increases with the layer hierarchy (Figure 6G), with the final layer achieving an accuracy exceeding 60%, significantly surpassing the random baseline accuracy of 10%.
>
> [1] Julesz, Bela. "Textons, the elements of texture perception, and their interactions." Nature 290.5802 (1981): 91-97.

---

> > ### Comment · Reviewer_moHy · 2023-08-19
> > **Reply to rebuttal**
> >
> > I appreciate the response and clarifications by the authors. I'm still confused about the joint/marginal generation though. For marginal generation, in principle don't you need to sample with O(n) time complexity conditioned on each z sample? Then the total time complexity would still be O(m*n)?

---

> > > ### Author Response · Authors · 2023-08-20
> > >
> > > We apologize if our previous explanation was unclear. Let me provide a clearer explanation.
> > >
> > > To generate an observation $\tilde{\mathbf{x}}$, we need to sample a pair $(\tilde{\mathbf{x}},\tilde{\mathbf{z}})$ from the distribution $p_\theta(\mathbf{x},\mathbf{z})$, where $\tilde{\mathbf{x}}$ represents the desired observation, such as an image.
> > >
> > > In informal terms, the process of sampling $(\tilde{\mathbf{x}},\tilde{\mathbf{z}})$ can be understood as searching for this specific pair $(\tilde{\mathbf{x}},\tilde{\mathbf{z}})$.  We assume that the size of the x-space is $O(m)$ and the size of the z-space is $O(n)$,
> > >
> > > In joint generation, when sampling the pair $(\tilde{\mathbf{x}},\tilde{\mathbf{z}})$, the search is conducted simultaneously in both the x-space and z-space, resulting in a required search space of $O(m*n)$. In marginal generation, when sampling the pair $(\tilde{\mathbf{x}},\tilde{\mathbf{z}})$, the process involves initially searching in the z-space according to $p_\theta(\mathbf{z})$. Once $\tilde{\mathbf{z}}$ is found, it is fixed. This step's search space size is $O(m)$. Then, in the x-space, $\tilde{\mathbf{x}}$ is searched based on $p_\theta(\mathbf{x}|\tilde{\mathbf{z}})$. This step's search space size is $O(n)$, leading to a combined required search space size of $O(m+n)$. When the sampling efficiency is comparable for both generation methods, the needed time is roughly proportional to the size of the search space.

---

### Official Review · Reviewer_tHDe · 2023-06-29

**Soundness:** 2 fair
**Presentation:** 1 poor
**Contribution:** 2 fair
**Rating:** 3
**Confidence:** 4

**Summary:**

This paper proposes a new model for sampling-based neural inference. The model posits that the brain attempts to match the marginal distribution $p(\mathbf{x})$ of an internal generative model $p(\mathbf{x}|\mathbf{z})$ (based on a neural representation $\mathbf{z}$) to the observed distribution of sensory inputs. In the model, sampling occurs via Langevin dynamics. The contributions of the paper are 1) use of a hierarchical exponential family model, 2) introduction of a second latent represenation $\mathbf{u}$ (putatively identified with interneuron dynamics) responsible for estimating the demeaned residual $\phi(\mathbf{x}) - A'(\boldsymbol{\eta})$, and 3) use of a second-order Langevin sampling scheme. Experiments show that the resulting generative model can perform on toy data sets and there are further claims about relation to neural data in the experiments.

This is a potentially interesting idea that I found to be muddled by lack of justification of several modeling choices and unclear exposition, particularly in discussing the rationale and results of the experiments.


**Strengths:**

- The neural sampling framework has been an interesting line of inquiry over the previous decade, particularly in thinking about how the brain might implement Bayesian inference, and this work furthers that approach.
- I found the use of a hierarchical model and the second latent representation to be interesting technical twists that are potentially powerful.
- The paper attempts to align both the additional latents and the second-order Langevin dynamics with known features of neural physiology.

**Weaknesses:**

- The paper is written as if its main contribution will be to theoretical neuroscience, but the experiments present evaluations of the generative model on toy data sets, and the experiments underlying Figure 6 have an unclear rationale.
- Overall, the paper has some deficits in presentation. For instance, section 2 is somewhat confusingly written. There is some parallel material that might be merged and better motivation given to the proposed choices. Why Langevin dynamics? Why not some other sampling method? Similarly, I didn't find the diagrams in Figure 1 particularly helpful in giving an intuition for the math.
- Likewise, the rationale behind most of the experiments is not clearly explained in the text, particularly, those underlying Figure 6. Captions are too sparse to explicate what the results mean, and the discussion in text only barely explains the relevant neuroscience background. I have no idea what analysis was done for Figures 6A and 6B, nor why. Figures 5 and 6E are impossibly small.
- ll. 120-23: This is a pretty compressed discussion of previously proposed approaches and perhaps difficult to follow for a non-expert reader.
- More generally, identifications of model constructs with neuroscience findings ($\mathbf{u}$ with interneuron activity, $\mathbf{v}$ with spike-rate adaptation) are simply stated without any real justification. I'm not saying these choices are impossible to justify, but no real arguments that these are plausible assumptions are given. Again, if this is supposed to be a generative model loosely based on neuroscience, that can be fine, but if it's a theoretical neuroscience model, it's not clear these choices have been fully thought-through, and the experiments should ideally be focused on demonstrating that the model can perform like the brain in some perceptual task.

**Questions:**

- ll. 152-156: It's clear from (17) that $\mathbf{v}$ represents a a noise term with temporal autocorrelation. In typical neural models, this is assumed semi-empirically as the result of temporal correlations among a large number of inputs to a given system. Is there an intuition the authors can give in this text for why it is better to identify (17) with spike frequency adaption as opposed to the existence of autocorrelated noise in synaptic inputs?
- It's not very clear from the results presented what benefits hierarchy in processing brings. Can the authors show in some more convincing way which posterior features hierarchy specifically helps to capture?

**Limitations:**

- The model in the paper assumes a feedforward generative model rather than a recurrent model, as in most brain circuits.
- The assumption $\tau_u \ll \tau_z$ (line 128) is a strong one, and one presumes is justified by the relatively higher firing rates of cortical interneurons compared with principal neurons. But can this be justified from firing rates alone? Is there an argument that needs to be made about relative equilibration times of neuronal voltage dynamics for these two types of cells to substantiate this?
- It's clear from the material in the supplement that some versions of the model cannot deal with multimodal posteriors, and even the second-order Langevin dynamics proposed will have difficulty mixing well in high-dimensional latent spaces.

---

> ### Author Rebuttal · Authors · 2023-08-09
>
> Thank you very much for your thorough review and valuable suggestions. We hope the following explanations will help alleviate the concerns raised in the weaknesses, questions, and limitations.
>
> **Weaknesses:**
>
> 1. As the reviewer correctly pointed out, our primary contribution is on theoretical neuroscience. Previously, Bayesian brain models have mainly focused on the inference process by using a pre-defined generative model and rarely considered how the generative model was learned. Here, we have taken a step further by explicitly modeling the learning process within the generative model framework. This inclusion may potentially give us insight into the learning mechanism in the brain's perceptual system. In Section 5.1, we employed several generative datasets to assess the learning capabilities of different energy models, and these evaluations help us to identify energy models of higher expression power that can learn complex data distributions. We used three datasets. Notably, CIFAR10 bears the closest resemblance to natural scenes the brain encounters. Therefore, we leveraged the representations learned from CIFAR10 to draw comparisons with neural representations in the brain in Section 5.2, and comparisons indicate that a HEE-based generative model may underlie the mechanism of the brain representing the external world information. This forms the rationale behind Section 5.2 and Figure 6.
>
> 2. We appreciate the feedbacks on our presentation, and we will improve Section 2, Figure 1, and all other parts of the paper for better clarity. The reason for we choosing Langevin dynamics as an example to introduce our overall framework in Section 2 is that: Langevin dynamics has been used in previous works to successfully model the sampling-based neural inference (see e.g., [1-2]), which helps readers to understand our framework easily (we will add this underlying rationale in the revised manuscript). In Section 4, we also discussed alternative sampling methods the brain might employ, such as second-order Langevin dynamics.
>
> 3. Thanks for the comments and suggestions. We will include the analysis details of Figure 6 and enlarge Figure 5 in the revised version. Figure 6 compares the learned representations of the HEE model from CIFAR10 (a natural image dataset) with the real brain representations in several facets, aiming to demonstrate the potential role of a generative framework in neural information representation.
>
> 4. We are grateful for the suggestions, and we will elaborate and integrate those short discussions on previous methods isolatedly appeared in several parts of the paper (line 34-38, line 120-23, and line 255-261).
>
> 5. As pointed out by the reviewer, our current work lacks a thorough exploration of the model's correspondence with the real neural system. Here, as the first step, our primary goal is to establish a framework, which addresses a particularly challenging problem in theoretical neuroscience, i.e., achieving simultaneous integration of biologically plausible local learning and sampling-based inference within a hierarchical network structure. But, as pointed out by the reviewer, we should at least give some justifications about the potential biological plausibility of model setting, which we will do definitely in the revised manuscript.
>
> **Questions:**
>
> 1. Thanks very much for the insightful suggestions. Indeed, interpreting eq.(17) as extracting the autocorrelation of inputs is biologically more reasonable. We will take this advice in the revised manuscript.
>
> 2. Previous researches have widely discussed about the advantages of hierarchical networks on information representation/learning. Compared to single or double-layer structures, deep networks can better approximate complex probability distributions p(x), enabling deeper layers to extract more abstract features [5], and these features further facilitate downstream tasks in the brain, such as decision making. In our work, Section 4 (lines 172-179) addresses the impact of network depth on convergence speed while keeping the total number of neurons fixed. Notably, an optimal network depth exists, as depicted in Figure 3D. Furthermore, our experiments demonstrate that as the number of layers increases, neural representations exhibit an enhanced linear discriminative capacity for object categories (Figure 6G), while neurons with orientation tuning decrease in number (Figure 6D). We will expand this discussion in the revised version.
>
> **Limitations:**
>
> 1. Please note that although our generative model adheres to a Markovian process and can be perceived as a feedforward process by graphical model (Figure 2A & Eq.(7)), the neural implementation of our generative model actually involves recurrent connections between neurons, resembling a recurrent neural network (Figure 2B & Eq.(9-10)). Modeling brain circuits involves neural networks rather than graphical model. This is also true for other generative models [1-4].
>
> 2. We appreciate the reviewer for offering two theoretical approaches that could potentially substantiate this hypothesis. Please see **rebuttal to all reviewers (part 2)** for further discussion.
>
> 3. In our framework, we need suitable energy models that can express multimodal distributions (as this is essential for capturing the statistics of natural images). Therefore, in Section 5.1, we assessed the expressive powers of different energy models and selected the one which is capable of representing multimodal distributions for the followed study.
>
> We are deeply grateful for the comments of the reviewer, which has provided significant insights for improving our work. We are committed to addressing these issues comprehensively in the revised version, and hope that the reviewer can raise the score accordingly.
>
> [1] Rodrigo Echeveste,Nature neuroscience,2020
> [2] Agnieszka Grabska-Barwinska, NeruIPS,2013
> [3] Nessler, B,Pfeiffer, PLoS Computational Biology,2013
> [4] Hennequin,NeurIPS,2014
> [5] DiCarlo,Neuron,2013

---

> > ### Comment · Reviewer_tHDe · 2023-08-10
> >
> > I appreciate the authors' responses to my concerns. While I do think there are interesting ideas here, I am concerned by the amount of revision that would be required to a) more fully situate this paper in a theoretical neuroscience context and b) perform appropriate comparisons of their model to similar approaches.

---

> > > ### Author Response · Authors · 2023-08-11
> > > **Revision and future work**
> > >
> > > Thank you for summarizing the improvements we need to make in the revised version. In order to address the reviewer's concerns and remind us to make a better revision to the paper, we outline the changes we are definitely planning to incorporate in the revision. Additionally, we detail the future work that will follow this study. We provide real-time updates on the progress in our **reply to all reviewers.**

---

### Official Review · Reviewer_nSfE · 2023-06-30

**Soundness:** 3 good
**Presentation:** 2 fair
**Contribution:** 4 excellent
**Rating:** 6
**Confidence:** 3

**Summary:**

In this article the authors introduce a new model, called the ‘’Hierarchical Exponential family Energy’ (HEE) that is biologically plausible and captures the dynamic of inference and learning in the brain. The model introduces multiple layers to decompose the EBM normalizing constant in a bio-plausible fashion, and leverages a neural adaption mechanism to make the sampling process compatible with ‘biological time’. In a series of experiments on synthetic dataset, Fashion MNIST and CIFAR10, the authors demonstrate the generation abilities of the HEE model. In addition, the authors demonstrate that the representation elicited by the HEE are similar to those observed in biological systems.

**Strengths:**

This article offers an interesting and valuable perspective on Energy Based Models (EBM) taking inspiration from neuroscience. The proposed solution to the intractable normalizing constant is original and elegant, and its link to predictive coding is interesting. Overall, there are plenty of good ideas in these article.


**Weaknesses:**

My biggest concerns are related to the experimental part:  (1) ablation studies that would validate the mathematical choice of the authors are missing , and (2) in general the experiments lack careful description. In addition, (3) the last part concerning the similarities between biological and HEE representation lack better comparison with other models. See more detailed comment below:


1 ) One of the innovation of this article lies in the decomposition of the intractable partition function using multiple layers. Intuitively I do see the advantage of the proposed method compared to the bufferization of the amortization method used in standard EBM. But I was expected a comparative experiment between the 3 different methods to confirm the advantage of the proposed method. The comparison with the IEBM of [1] is not carefully enough conducted to conclude to the superiority of the proposed method : the parametrization seems to be different (CNN versus FullyConnected + number of parameters) and other improvement are introduced (e.g. neural adaptation) that might biased the comparison. In general I would suggest more ablation experiments that would test the benefits of all the HEE components separately.

2) The networks are poorly described in the experimental part. What is the exact parametrization of the network (dimension of the fully connected network) ? What is the exact sampling procedure you use in the HEE : Do you first wait for a stable point in the sampling of the latent variable x_l before sampling the next one ? Or do you do one step of sampling for all the variables until you reach a global stable point ? Such implementation details might lead to very different equilibrium state and should be at least discussed. Also, how many sampling steps are you using ? Or are you forcing a stoping criterion (based on the prediction error ?) to decide when to stop the sampling process ?

3) The experimental results in section 5.2 lacks comparison with other similar model. There is nothing showing the results you show are not  general to all networks (and not specific to HEE). Could you compare the HEE representations with those obtained for other EBM (or even other generative model) ? The idea would be to demonstrate that an EBM compatible (in terms of implementation) with biology would produce representation more aligned with biology.


**Questions:**

* You did not include any comparison with other EBM for Fashion MNIST or the synthetic dataset. Is there a reason for that ? The synthetic dataset would have offered the perfect opportunity to conduct the comparison described previously .

* One of the major difficulty in the IEBM [1], is the lack of stability of the training procedure (partly due to the mode traversing issue in the negative phase). Have you faced the same issue with the HEE ?

* Line 200 : I don’t understand the sentence : "The dropout technique is employed as a method to mimic the receptive field behavior found in biological systems. » Could you further explain ?

* Have you tried some convolutional architecture ? Don’t you think it would improve the results ?

* The difference between HEE-NL-A and HEE-NL is only the number of layers ? This should be clarified.

* In Table 1, could you define LS and SLD ?

* In Fig5, the samples seems be visually lower quality than those of the IEBM [1], but the FID and IS are better. Do you have an explanation ? In particular, in the samples you present, the textures seem to be rather uniform…

* What is the impact of the number of layers in the quality of the generation ??


Typo :
Line 65 : EMB —> EBM
Line 135: PCN is used but not defined yet.

[1] Du, Yilun, and Igor Mordatch. "Implicit generation and generalization in energy-based models." arXiv preprint arXiv:1903.08689 (2019).

**Limitations:**

Limitations have been properly addressed by the authors

---

> ### Author Rebuttal · Authors · 2023-08-09
>
> We acknowledge the encouraging and valuable comments of the reviewer, and would like to address the concerns of the reviewer in details below.
>
> **Weaknesses:**
> 1. Please see **rebuttal to all reviewers (part 3)**.
>
> 2. We recognize the current paper lacks detailed experiment descriptions, and we commit to add them in the revised manuscript. We will also incorporate a diagram illustrating the neuronal connections and network parameters.
>
>     The network parameters, denoted as $\theta_l$, represent the connection weights between $x_{l+1}$ and $\varepsilon_l$ (dashed arrows in Figure 2B). In the experiment on CIFAR-10 dataset , 80% of $\theta_l$ elements are randomly set to zero permanently, effectively disconnecting 80% of connections between $x_{l+1}$ and $\varepsilon_l$. We refer to this operation as "dropout" following the terminology in machine learning, but perhaps “sparse connectivity” is a better word.
>
>     Our sampling approach involves performing every step of sampling for all variables till a global stable point is reached. This approach aligns better with the biological plausibility, and we have yet explored hierarchical sampling. We have presented the details of the training and generation processes in Supplementary Material (section 3). During training, we stop sampling after every 300 $\tau_x$. time steps, using the Euler method with a time step of $dt = 0.01\tau_x$.
>
> 3. We appreciate the suggestion and plan to incorporate these experiments and results in the revision. We are considering to add data from IEBM and PixelCNN [1]. Additionally, in future work, we intend to compare our model's results with the neural prediction accuracy benchmarks [2], by using our model's representations to predict the real brain neuron representations.
>
>     [1] Aaron Van Oord, Nal Kalchbrenner, and Koray Kavukcuoglu. Pixel recurrent neural networks. In ICML, 2016.
>
>     [2] Zhuang, Chengxu, et al. "Unsupervised neural network models of the ventral visual stream." Proceedings of the National Academy of Sciences 118.3 (2021): e2014196118.
>
> **Questions:**
>
> 1. Because our primary focus is on theoretical neuroscience, our work aims to present a biologically plausible learning method. The purpose of the experimental section is to demonstrate that our proposed model (a generative model) could potentially underlie the mechanism behind perception in the brain. The experiments conducted on the 2D synthetic and FashionMNIST datasets in Section 5.1 were primarily intended to showcase the learning capabilities of our proposed 'local learning' approach. We aimed to compare different sampling methods and identify models with favorable performance. Subsequently, the study of using the CIFAR10 dataset was motivated by its inclusion of natural images that closely resemble visual inputs received by the brain. This evaluation aimed to provide evidence that our model indeed learns the underlying distribution of these natural images. This serves as a foundation to support the discussion in Section 5.2, where we explored the model's representations and their alignment with real brain neuron representations. In the revised manuscript, we will include performance comparisons with IEBM on 2D synthetic datasets and the FashionMNIST dataset.
>
> 2. While we have not trained IEBM, we can't pinpoint the exact reason for the issue it encountered. However, we did face instances of sudden gradient increasing and model instability during the training. After careful analysis, we identified that the uneven data distribution during learning led to inflated eigenvalues of the Jacobian matrix in the sampling dynamic Eq. (10). To mitigate this, we introduced a regularization term to the function $g(x)$, as described in Supplementary Material (section 3 and $g(x)$ in Table 1).
>
> 3. We randomly set 80% of $\theta_l$ elements to zero, effectively disconnecting 80% of connections between $x_{l+1}$ and $\varepsilon_l$. This implies that during inference, each neuron in the second neural network layer only receives 20% of information from the first layer directly, and this setting extends to subsequent layers (each neuron in the third layer can receive information from the first layer equivalent to 1 - 0.8 * 0.8.). This concept closely resembles the receptive field concept in the visual system.
>
> 4. We have not attempted this approach, as it would compromise biological plausibility (weight sharing is not biologically plausible). To better compare with IEBM, we plan to explore the CNN structure in future work.
>
> 5. HEE-NL-A and HEE-NL differ in their sampling methods. HEE-NL employs Langevin sampling without introducing adaptation, while HEE-NL-A incorporates second-order Langevin dynamics for sampling.
>
> 6. LS stands for Langevin sampling, and SLD for second-order Langevin dynamics. We will provide detailed explanations in Table 1 in the revised manuscript.
>
> 7. Our IS (6.47) is indeed less than the 10-ensemble approach of IEBM (6.78), while utilizing FID for evaluating our results yield better. Simply speaking, IS measures the quality of generated images, while FID assesses the similarity between generated images and those within the dataset. Therefore, in term of scoring, our generated image quality might not be as high as IEBM's, but the similarity of our generated images to CIFAR10 is closer compared to IEBM. Visually, the perceived lower quality and uniform texture might be attributed to the small size of Figure 5. Additionally, the images not being vector graphics could contribute to this. In the revised version, we will address this concern by including larger generated images in the Supplementary Material. This will provide a better opportunity to observe the finer details of generated images.
>
> 8. For FashionMNIST, we tried L=5, 10, 15, and 30. L=5 yielded the worst result, L=10 and L=15 exhibited similar performance, L=30 faced gradient explosion during training. Hence, we employed L=10 on the CIFAR10 dataset.

---

> > ### Comment · Reviewer_nSfE · 2023-08-21
> > **Response to authors**
> >
> > Sorry for the late feedback. I appreciate the detailed answer from the reviewer. I can't really update my rating without seeing a proper (and well-controlled) comparison between the proposed method and the IEBM... I think this paper would be greatly improve if the authors can include such a comparison (and also more details concerning the experiment part).

---

> > > ### Author Response · Authors · 2023-08-21
> > >
> > > As we are unable to submit the revised article at this moment, we can only provide a brief report here on the comparison with IEBM after controlling for more variables.
> > >
> > > **CIFAR-10 Unconditional:**
> > >
> > > | Model  | IS | FID | Sampling method | Network structure | Parameters |
> > > |  :----:  | :----:  |  :----:  | :----:  | :----:  | :----:  |
> > > | IEBM  |  6.78 | 38.20 | Langevin | ResNet | 5M|
> > > | HEE (previous)   |  6.47  | 37.05 | second-order Langevin | Fully connected | 4M |
> > > | HEE (controlled)  |  7.07 | 33.37 | Langevin | CNN (without skip connections) | 4M |
> > > | HEE (controlled)  |  7.12 | 32.10 | Langevin | CNN (without skip connections) | 5M |
> > >
> > > We found that the CNN structure not only produces higher quality in terms of generation but also reaches steady state in a shorter time. During training, the fully connected structure requires $300\tau_x$, whereas the CNN structure only requires $100\tau_x$. When generating images, the fully connected structure requires $100\tau_x$, while the CNN structure only needs $50\tau_x$.
> > >
> > > We will provide a more detailed report of this result in the revised version.

---

### Official Review · Reviewer_zTHc · 2023-07-05

**Soundness:** 3 good
**Presentation:** 1 poor
**Contribution:** 3 good
**Rating:** 5
**Confidence:** 3

**Summary:**

In this submission, the authors introduce a biologically plausible method to train a hierarchical energy-based model as well as perform inference over it, via Langevin sampling. This accomplishes a goal that has long been sought in computational neuroscience, namely, neurally plausible methods for inference over a hierarchical generative model of the world.

Previously, the main problem facing inference in hierarchical energy-based models is the difficulty of evaluating the partition function. The authors address this issue by modeling the partition function with Langevin sampling, but with a faster timescale than the Langevin sampling used for inference more generally. This appears to work well empirically. Also underlying their model's impressive generation performance is the use of a momentum term in the Langevin sampling, similar to Hamiltonian Monte Carlo, which they associate with short-term adaptation in the brain. Learning in this system is Hebbian and biologically plausible, and additionally the authors show some rough parallels with biological phenomena, such that the inferred activations of selected neurons have orientation- and hue-specific tuning curves.

**Strengths:**

I was impressed with the quality of samples generated by this hierarchical EBM, and all the more so because the learning process is so simple and Hebbian in nature. The model aligns well with a wide literature in computational neuroscience, including hierarchical predictive coding networks, and it extends previous EBM models in an important way by enabling a pure sampling-based inference procedure. The model combines multiple previous ideas into a single framework, and the approach feels natural.

**Weaknesses:**

Although generation was impressive, this is also a proposal for Bayesian inference, and there were no serious benchmarks quantifying inference. Some thought should be put into evaluating known benchmarks for inference over hierarchical energy-based models. Many examples are possible, e.g. hierarchical gaussian mixture models, etc.

Clarity was a major issue. This was difficult to read. If this is going to be widely read, it should to be rewritten. I’ll describe specific problems below.

First, there was not sufficient context in the surrounding literature. The introduction is rather short about EBMs. Otherwise it is difficult to know about this work’s specific contribution as many of these ideas have separately been introduced before (H-EBMs, Hamiltonian MC, etc.).  Also, for example, section 2 should summarize previous hierarchal EBMs, as currently it reads like the authors’ original proposal (which it is not). The related work section should be greatly expanded.

There were not also enough details about the experimental methods. I would certainly not be able to reproduce these figures from the manuscript. If there is not enough room, at least please put more details in the supplementary figures.

For clarity, I would recommend adding more detail about the biological circuit proposal. It is difficult for me to imagine the connectivity of the inhibitory neurons. (Do they need to project from one area to another, for example?) A figure with biology in mind would help. More details about biology should be added, as right now it is rather vague as to how this would be implemented in the brain.

In addition, many sentences state a controversial hypothesis as a known fact. These should be weakened considerably. For example,
Line 46: “In this study, we show that our brain holds an intrinsic energy-based model”. ‘Show’ implies a certain conclusion. Better would be ‘hypothesize’ or ‘propose’.
Line 70: “We demonstrate that our brain holds an intrinsic energy-based model”. Again, ‘demonstrate’ is conclusive. You are demonstrating that the brain *might* use such a model.
Line 1: “The brain engages in probabilistic inference” This is just a hypothesis, although a popular one. Neural networks can appear to be Bayes-optimal even though they are not at all probabilistic internally. See Orhan, A. Emin, and Wei Ji Ma. "Efficient probabilistic inference in generic neural networks trained with non-probabilistic feedback." Nature communications 8.1 (2017): 138.


**Minor comments:**

The word ‘society’ is used when I think the authors want ‘community’.

In general the sentences are often awkwardly phrased. I recommend asking for editing help, perhaps using a large language model for grammar.

Section 4. This seems to be Hamiltonian Monte Carlo. Is that correct? If so, this should be cited (e.g. Neal (2010)) as well as previous HMC in the brain proposals, like the Aitchison and Lengyel (2016) paper cited elsewhere. If it is incorrect, at least please cite SLD.

Line 31: Many have argued that variational methods are biological plausible. See for example:
D. Rezende and W. Gerstner, “Stochastic variational learning in recurrent spiking networks,” Frontiers in Computational Neuroscience, vol. 8, p. 38, 2014, doi: 10.3389/fncom.2014.00038.

Line 84: “avoid complex calculations”. Could you be more specific?
Line 121-124. This sentence is unclear and needs to be unpacked. What is amortized generation method, etc.


**Questions:**

 - How many long does convergence take to produce the samples shown in the manuscript?
 - What is the connectivity of the interneurons? What cell types might these be? What evidence is there for their faster time constants?
 - I don't fully understand the circuit diagram for biological neurons. Could a schematic be drawn with individual neurons, complete with a table of the key properties of those neurons in this theory?

**Limitations:**

There was not much mention of any limitations of this method. What are the least biologically plausible aspects? What aspects need to be experimentally confirmed?

---

> ### Author Rebuttal · Authors · 2023-08-09
>
> We acknowledge the encouraging and valuable comments of the reviewer, and would like to address the concerns of the reviewer in details below.
>
> **Weaknesses:**
>
> a) Quantifying Inference:
> Thank for the suggestion of incorporating benchmarks into the inference section. This aligns with our future plan. Besides the benchmarks you mentioned, we intend to compare our results with "neural prediction accuracy" benchmarks [1]. This benchmark entails using our model's representations to predict real brain neuron representations.
>
> [1] Zhuang, Chengxu, et al. "Unsupervised neural network models of the ventral visual stream." Proceedings of the National Academy of Sciences 118.3 (2021): e2014196118.
>
> b) Clarity:
> Thank for the suggestions, in the revised version, we will
>
> 1. provide a more comprehensive introduction to EBMs to ensure a clear understanding of our work.
>
> 2. largely re-write the figure 6 caption to provide detailed descriptions for each subfigure.
>
> 3. merge figures 1 and 2 in the revision and present the neural connection diagram.
>
> 4. improve the writing and we will diligently address the parts highlighted by the reviewer, alongside other instances.
>
> c) Minor Comments:
>
> In Section 4, the second-order Langevin sampling we used indeed bears similarity to Hamiltonian Monte Carlo, although they are not entirely the same. Both methods fall within the broader family of sampling techniques involving auxiliary variables to accelerate the sampling process [1]. Neal (2010) first introduced the concept of augmenting sampling speed using auxiliary variables, and Aitchison and Lengyel (2016) extended this to neuroscience by employing a group of inhibitory neurons as extra variables. In our work, we propose using adaptation current as an auxiliary variable. We will appropriately cite the references in the revised version.
>
> As pointed out by the reviewer, the potential application of variational methods in the brain have been explored before. Some studies have also compared variational methods with sampling-based methods to determine their relative merits [2]. Sampling-based methods face convergence challenge, while variational methods involve both inference and generation stages during training, posing difficulties in multi-layered structure. In our current work, we primarily discussed the potential of sampling-based methods. Determining the brain's exact strategy requires further investigation, and we will duly address variational methods in the revised version.
>
> The inference process of a generative model involves calculating the posterior distribution $p(z|x)$, where $p(z|x)=p(x,z)/p(x)$. Computing the denominator term, $p(x) = \int p(x,z) dz$, becomes challenging when the dimensionality of $z$ is high. This integral can lead to complex calculations. Variational methods circumvent this by approximating $p(z|x)$ using a tractable distribution $q(z|x)$, thus avoiding the need to calculate $p(x)$. In sampling methods, like ours, we only require knowledge of $\nabla_z \ln p(z|x)$ to obtain samples that follow the distribution $p(z|x)$. Considering $\nabla_z \ln p(z|x)= \nabla_z \ln p(x,z)$, we also avoid the need to calculate $p(x)$.
>
> [1] Ma, Yi-An, Tianqi Chen, and Emily Fox. "A complete recipe for stochastic gradient MCMC." Advances in neural information processing systems 28 (2015).
>
> [2] Grabska-Barwinska, Agnieszka, et al. "Demixing odors-fast inference in olfaction." Advances in Neural Information Processing Systems 26 (2013).
>
> **Questions:**
>
> a) Convergence speed depends on network size and depth. For our CIFAR10 dataset, with a network of 10 layers and a total of 150k neurons, typically 50-100 $\tau_z$ steps are needed for high-quality image generation.
>
> b) Regarding Figure 2B, interneurons $\varepsilon_l$ and principal neurons $x_l$, $x_{l+1}$ are connected. The $\varepsilon_l-x_l$ connections follow one-to-one pairing, and $\varepsilon_l-x_{l+1}$ connections are established via the weight matrix $\theta_l$. The internal connections of $\varepsilon_l$ are determined by $g'(u_l)$ as described in Eq.(9). The discussion of interneurons is addressed in the **rebuttal to all reviewers (part 2)**.
>
> c) In the current manuscript, it is not possible to discern the specific neuronal connections from the figures alone. One would need to refer to the corresponding dynamic equations Eq.(9-10) to understand the precise connections. We will address this shortcoming in the revision by providing an illustration of the neuronal connectivity diagram.
>
>
> **Limitations:**
>
> *“What are the least biologically plausible aspects? ”*
>
> As you and many other reviewers have pointed out, the assumption $\tau_u \ll \tau_z$ (line 128) is indeed a strong one. The rationale behind making this assumption is that during the inference process described by Eq. (8), we need to have real-time knowledge of the value of $A'(\eta_l)$. However, we compute $A'(\eta_l)$ through sampling (as in Eq. (9)), and sampling inherently introduces some temporal delay. To minimize this delay, we need to ensure that the timescale of sampling $A'(\eta_l)$, denoted as $\tau_u$, is much smaller than the timescale of inference dynamics $\tau_z$. Nevertheless, we have also identified some empirical evidence that supports the validity of this assumption in the reply to all reviewers.
>
> *“What aspects need to be experimentally confirmed?”*
>
> 1.Our model assumes SFA as the adaptation term to accelerate the sampling process (as discussed in Section 4). Suppose one can manipulate the SFA strength in the experiment, then we can test this assumption.
>
> 2.Additionally, our model offers substantial flexibility, including the choices of $\phi(x)$, $\eta_l$, and $g(x)$. Currently, our selections primarily consider their impact on the model's learning capability (as discussed in Section 5.1), without necessarily aligning with real neural systems. Perhaps in the future, we could conduct experiments to validate the appropriateness of these choices.

---

> > ### Comment · Reviewer_zTHc · 2023-08-10
> > **Thanks for the detailed response**
> >
> > I'm happy to see that all reviewers engaged thoroughly, and that the authors are taking all such feedback seriously. If the authors do seriously implement all of the proposed changes, I think this would be a valuable paper at NeurIPS. Without actually seeing this revision, though, I cannot in good faith yet change my score. However, I want to emphasize that my criticisms are not about the soundness of the proposed method, but only about its contextualization, clarity, the depth of explicit relation to biology, and overall presentation, all of which can (in principle) be addressed in revision.

---

> > > ### Author Response · Authors · 2023-08-11
> > > **Revision and future work**
> > >
> > > Thank you for acknowledging our work. In order to address the reviewer's concerns and remind us to make better a revision to the paper, we outline the changes we are definitely planning to incorporate in the revision. Additionally, we detail the future work that will follow this study. We provide real-time updates on the progress in our **reply to all reviewers**.

---

### Author Rebuttal · Authors · 2023-08-09

**Reply to all reviewers**

We greatly appreciate the thorough review of our paper by all four reviewers. The positive feedbacks in the reviews have effectively highlighted the contributions of our work. We will provide a summary of this aspect in Part 1. Additionally, we will address the common questions raised about our work in Parts 2 and 3.

**Part 1**

Previously, Bayesian brain models primarily emphasized the inference process, utilizing pre-defined generative models, and rarely delved into how the generative model was acquired. In this work, we have extended beyond this by explicitly incorporating the learning process within the hierarchical generative model framework. This addition could potentially offer insights into the learning mechanism within the brain's perceptual system.

Furthermore, our brain-inspired energy-based model (EBM) also presents a technique for estimating the partition function in EBMs, which is a challenging problem within the machine learning community.

**Part 2**

As reviewers zTHc, tHDe and moHy pointed out, $\tau_u\ll\tau_z$ (line 128) is a strong assumption. In the current paper, we do acknowledge the absence of a discussion on this assumption and interneurons. Therefore, we are providing additional discussion here and will incorporate this aspect into the revised version.

There are various types of interneurons that target on pyramidal cells, comprising approximately 10-20% of the overall neuron population in the cerebral cortex [1]. The interneurons in the HEE model bear the closest resemblance to the Large Basket Cell or Nest Basket Cell [2], which collectively constitute around 50% of interneurons. Their electrophysiological characteristics include fast spiking, non-accommodating, and non-adapting behaviors. These interneurons have also been identified in the visual cortex of ferrets [3], displaying short-duration action potentials (approximately 0.5 ms at half height). This suggests that these neurons have shorter time constants compared to pyramidal cells.

[1] Therese Riedemann. Diversity and function of somatostatin-expressing interneurons in the cerebral cortex. International Journal ofMolecular Sciences, 20(12):2952, 2019.

[2] Markram, Henry, et al. "Reconstruction and simulation of neocortical microcircuitry." Cell 163.2 (2015): 456-492.

[3] Descalzo, Vanessa F., et al. "Slow adaptation in fast-spiking neurons of visual cortex." Journal of neurophysiology 93.2 (2005): 1111-1118.

**Part 3**

As the reviewer nSfE pointed out, it is better to compare our model with other methods on the partition function estimation. However, we find that for the joint distribution in our model (belonging to the hierarchical exponential-family), implementing amortized generation and implicit generation are challenging. The difficulty arises from the fact that, in our model, besides requiring the partition function during training (Eq. 11), we also need the partition function during sampling (Eq. 8). In contrast, amortized generation and implicit generation do not consider partition function during the sampling process. Thus, if comparing our model with the other two methods, the joint distribution employed by them will be different from the one used in our model, and this makes comparison not in the same condition. We need carefully design experiments to address this issue.

As the reviewer pointed out, our comparison with IEBM lacks controlled variables. In term of connectivity, IEBM employs a CNN architecture, while our model employs a full connection structure. While our model could also be adapted to a CNN-like structure, the operation of weight sharing in CNN is not biologically plausible, as it requires the changes of synapses over a large space are coordinated. Therefore, in our model implementation, we chose a fully connected architecture instead of CNN to maintain better biological realism. Regarding the sampling method, IEBM employs Langevin sampling, while our approach employs Langevin sampling + adaptation. This technique, while not identical to Hamiltonian Monte Carlo (HMC), falls within the same category of sampling methods [1]. Notably, IEBM also discusses the impact of HMC and highlights the challenge in controlling the number of leapfrog simulations during training. In term of parameter count, IEBM utilizes around 5 million parameters, while our model employs 4 million. Additionally, in term of loss function, our approach directly employs log-likelihood, while IEBM incorporates an additional regularization term. Moving forward, in order to enhance the comparison with IEBM, we are considering a strategy that temporarily sets aside biological constraints, rather we will focus solely on experimental outcomes by adopting a CNN architecture, employing Langevin sampling, and ensuring parity in parameter count. By this, we can conduct a direct and informative comparison with IEBM.

[1] Ma, Y. A., Chen, T., & Fox, E. B. (2015). A complete recipe for stochastic gradient MCMC. Advances in Neural Information Processing Systems, 2015-Janua, 2917–2925.

---

> ### Author Response · Authors · 2023-08-11
> **Revision and future work**
>
> In order to address the reviewers' concerns and remind us to make better revisions to the paper, we outline the changes we are definitely planning to incorporate in the revision. Additionally, we detail the future work that will follow this study.
>
> **Revision:**
>
> a) Presentation
> 1. Provide additional discussion on interneurons. (suggested by zTHc, tHDe and moHy; done)
> 2. Provide more details for the figure 6 caption, a more comprehensive introduction to EBMs in Section 2. (zTHc,tHDe; done)
> 3. Present the neural connection diagram, rephrasing the inappropriate wording, cite Neal (2010) and Aitchison and Lengyel (2016) and discuss the variational method. (zTHc; done)
> 4. Change the term 'drop out' to 'sparse connections', include larger generated images in the Supplementary Material. (nSfE,tHDe; done)
> 5. Make a discussion on adaptation, interprete eq.(17) as extracting the autocorrelation of inputs and clarify the motivation of Sec.5.1 (tHDe; done)
> 6. Distinguish random variables and samples, show the difference between marginal and joint generation (moHy; done)
>
> b) Experiment
> 1. By adopting a CNN architecture, employing Langevin sampling, and ensuring parity in parameter count, conduct a direct and informative comparison with IEBM. (nSfE; done)
> 2. Quantifying inference results with IEBM and PixelCNN (nSfE, zTHc; done)
> 3. Include performance comparisons with IEBM on 2D synthetic datasets and the FashionMNIST dataset. (nSfE; done)
>
> **Future work:**
> 1. Design experiments to compare amortized generation, implicit generation and our methods. (nSfE; challenging)
> 2. Add 'neural prediction accuracy' benchmarks (nSfE, zTHc; done)
> 3. Conducting relevant experiments on stimuli with serial dependence (moHy; about to complete)

---

### Decision · Program_Chairs · 2023-09-21

**Decision:**

Accept (poster)

**Comment:**

This is a borderline case. Three reviewers voting for acceptance (with scores 5,6,8) , one for rejection (score 3). The main arguments for rejection are that the paper needs to be revised to more fully situate this paper in a theoretical neuroscience context. Reviewers however beliefe that a thoroughly revised draft can address all of the outstanding issues and that the theoretical novelty proposed in this article is an interesting contribution. I encourage the authors to incorporate all the promised changes into the revised version of the manuscript.